# Learning Adaptive Parallel Reasoning with Language Models

**Jiayi Pan**[*1], **Xiuyu Li**[*1], **Long Lian**[*1], **Charlie Snell**[1], **Yifei Zhou**[1],
**Adam Yala**[1,2], **Trevor Darrell**[1], **Kurt Keutzer**[1], **Alane Suhr**[1]
[1]UC Berkeley    [2]UCSF
{jiayipan,xiuyu,longlian,suhr}@berkeley.edu

## Abstract

Scaling inference-time computation has substantially improved the reasoning capabilities of language models. However, existing methods have significant limitations: serialized chain-of-thought approaches generate overly long outputs, leading to increased latency and exhausted context windows, while parallel methods such as self-consistency suffer from insufficient coordination, resulting in redundant computations and limited performance gains. To address these shortcomings, we propose Adaptive Parallel Reasoning (APR), a novel reasoning framework that enables language models to orchestrate both serialized and parallel computations end-to-end. APR generalizes existing reasoning methods by enabling adaptive multi-threaded inference using `spawn()` and `join()` operations. A key innovation is our end-to-end reinforcement learning strategy, optimizing both parent and child inference threads to enhance task success rate without requiring predefined reasoning structures. Experiments on the Countdown reasoning task demonstrate significant benefits of APR: (1) higher performance within the same context window (83.4% vs. 60.0% at 4k context); (2) superior scalability with increased computation (80.1% vs. 66.6% at 20k total tokens); (3) improved accuracy at equivalent latency (75.2% vs. 57.3% at approximately 5,000ms). APR represents a step towards enabling language models to autonomously optimize their reasoning processes through adaptive allocation of computation.

## 1 Introduction

Recent progress in language model reasoning, like OpenAI o1 (OpenAI, 2024) and DeepSeek-R1 (DeepSeek-AI, 2025), has demonstrated the promise of exploiting test-time compute to perform search and of using reinforcement learning to optimize the search. However, these current approaches face fundamental limitations: serialized chain-of-thought methods (DeepSeek-AI, 2025) produce lengthy output sequences that increase latency and strain context window limits, while parallel methods like best-of-N or self-consistency (Wang et al., 2023) often lack coordination between inference paths and are not end-to-end optimized, leading to redundant computation and limiting improvement. Structured inference-time search methods like tree-of-thought (Yao et al., 2023) require hand-designed search structures, limiting their flexibility and scalability.

We propose Adaptive Parallel Reasoning (APR), a simple yet effective approach that enables language models to reason by adaptively distributing inference-time computation in a manner that exploits both serial and parallel operations. Our method generalizes existing approaches to reasoning with language models, including serialized chain-of-thought reasoning, parallelized inference with self-consistency, and structured search: rather than imposing fixed search structures through prompting or external orchestration, we train language models to *learn* when and how to parallelize their inference operations.

Adaptive Parallel Reasoning employs two key innovations: First, we supply language models with a parent-child threading mechanism. Parent inference threads can, at any point during decoding, delegate subtasks to multiple child inference threads using a `spawn()`

---

*Equal contribution. Code, model, and data are available at github.com/Parallel-Reasoning/APR.

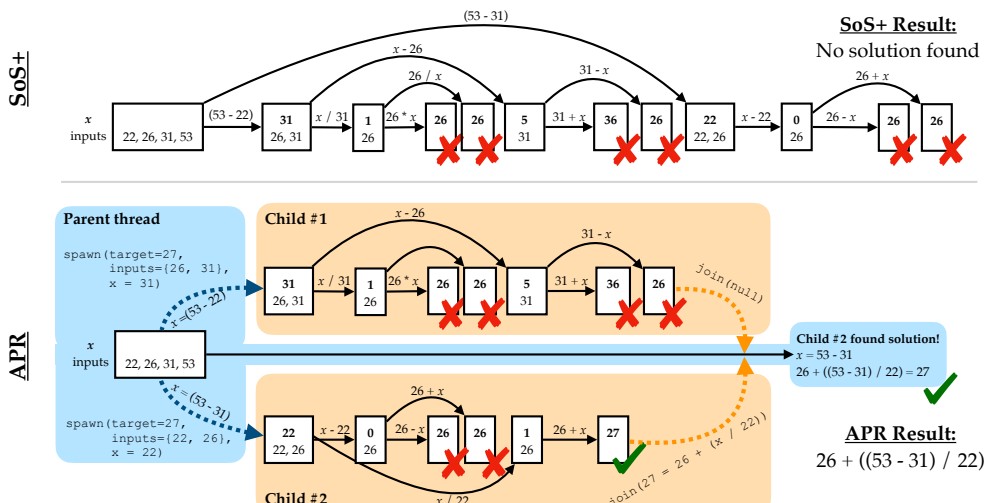

Figure 1: **Serialized search (Gandhi et al., 2024) (top) vs Adaptive Parallel Reasoning (bottom)** illustrated on an example of the Countdown task, with a target number of 27 and input numbers of {22, 26, 31, 53}. Each box represents a node in the search representing the value of the intermediate expression ($x$). Edges are annotated with explored arithmetic operations relative to the parent-node $x$ using remaining input numbers. In serialized search, the context window of the single inference thread is exhausted before a solution is found. In Adaptive Parallel Reasoning, the parent thread (blue) spawns two child threads (orange), which are executed in parallel. Child threads have access only to a limited context passed to them by the parent thread and return a summary of their execution to the parent thread. The parent thread can then continue to decode with access to these summaries. This parallel distribution of computation prevents context window exhaustion while reducing latency.

operation. Child threads independently explore distinct reasoning paths in parallel and return outcomes to the parent thread through a join() operation. The parent thread then continues decoding conditioned on the information returned by the children. We build on the model serving framework SGLang (Zheng et al., 2024) to perform inference in child threads simultaneously through batching, which significantly reduces real-time latency. Our second key innovation is to fine-tune the language model that performs inference in both parent and child threads via end-to-end reinforcement learning, which optimizes overall task success and eliminates the need to predefine explicit reasoning structure.

Figure 1 illustrates how APR facilitates more efficient and effective reasoning in the Countdown reasoning task (Yao et al., 2023; Gandhi et al., 2024; Pan et al., 2025) when compared to serialized methods. Our experiments demonstrate that we achieve three key benefits over prior approaches:

1. **Higher performance within same context window**: Our approach performs more effective search and reasoning within fixed context size constraints (83.4 vs 60.0% at 4k context) compared to sequential methods that quickly exhaust available context.

2. **Superior scaling behavior**: When scaling the total compute budget, APR exhibits better performance improvements (80.1 vs 66.6% with a budget of 20k total tokens) through wider parallelization in addition to increasing the length of individual reasoning chains.

3. **Improved performance at the same latency**: Adaptive Parallel Reasoning achieves significantly higher success rates compared to serialized search methods with same latency (75.2 vs 57.3% at around 5,000ms).

These results highlight the potential of training language models to adaptively allocate their own inference-time compute resources. By learning when to reason serially and when to branch out into parallel computation, models can more efficiently explore the solution space of complex reasoning problems.

## 2 Related Work

**Inference-time scaling** Prior work has shown that increasing test-time compute can improve language model performance on downstream tasks (Wei et al., 2022; Zelikman et al., 2022; Zhu et al., 2024; DeepSeek-AI, 2025; OpenAI, 2024; Team, 2025; Gandhi et al., 2024; Snell et al., 2025; Li et al., 2025). However, these methods typically result in significantly longer output sequences, introducing key limitations given the inherently sequential nature of autoregressive decoding: longer output sequences mean higher latency, and fitting an entire sequence into a single context window makes it hard for models to attend to relevant information when scaled with more output tokens. Compared to serialized inference, parallelizing reasoning traces reduces both latency and the pressure of context window limitations. Our experiments demonstrate that models trained end-to-end to adaptively distribute inference-time compute can outperform serialized approaches under the same computational budgets.

**Parallelization in language model inference** Parallelizing inference with multiple independent language model calls has emerged as another avenue for inference-time scaling towards improved reasoning performance (Cobbe et al., 2021; Wang et al., 2023). While these ensembling methods effectively enhance performance, their lack of coordination across parallel threads results in redundant computation and suboptimal resource utilization. Recent methods have aimed to address this limitation by proposing fixed parallelizable reasoning structures (Yao et al., 2023; Du et al., 2023; Kim et al., 2024; Grand et al., 2025; Schroeder et al., 2024; Ning et al., 2024; Zhang et al., 2024; Hua et al., 2024; Zhuge et al., 2024; Teng et al., 2025; Wang et al., 2025) often without learning. However, these proposed search structures (and, in the case of multi-agent reasoning, the communication protocol) are fixed and hand-designed, limiting their flexibility and scalability. In contrast, our approach leverages reinforcement learning to train language models to adaptively structure search at inference time, dynamically allocating compute between parallel and serial operations. Concurrent to our work, PASTA (Jin et al., 2025) and Hogwild! Inference (Rodionov et al., 2025) also explore parallel reasoning. However, PASTA decomposes tasks into parallel sub-tasks but ultimately merges the complete context from each sub-task back into the main inference trajectory, thus not effectively reducing context usage. Meanwhile, Hogwild! Inference employs parallel worker threads for collaborative reasoning, yet it relies exclusively on prompting without any end-to-end optimization. In contrast, our method uniquely integrates supervised training and reinforcement learning, enabling language models to adaptively and efficiently manage parallel explorations and selectively integrate successful outcomes, maximizing reasoning performance within constrained context windows.

Many parallel inference methods have shared prefixes across language model calls, which synergize with modern serving framework to improve serving efficiency via prefix caching and reuse (Zheng et al., 2024). APR builds on these optimizations, as parent-child threading mechanism creates natural opportunities for prefix sharing, which further reduces the computational overhead of APR parallelization.

**Training language models to control their own outputs** Our work is related to algorithms for training language models to directly control the inference process, for both efficiency and performance reasons. For example, PENCIL (Yang et al., 2025) trains models to adaptively discard parts of their context at inference time to reduce the context length, and thus the time complexity of producing a successful reasoning trace. Goyal et al. (2024) train language models to inject additional "pause" tokens into their output sequences, which are shown to improve end-task performance by increasing inference-time computation. Both approaches train using supervised demonstrations of token discarding (or injection). To the best of our knowledge, we are the first to use end-to-end reinforcement learning to optimize language model control over the allocation of inference compute.

## 3 Adaptive Parallel Reasoning

We now introduce Adaptive Parallel Reasoning (APR), which teaches language models to adaptively orchestrate parallel inference processes. By distributing compute across parent and parallel child threads, APR minimizes the end-to-end latency, achieves better performance within the same context limit constraint, and scales better with an increasing

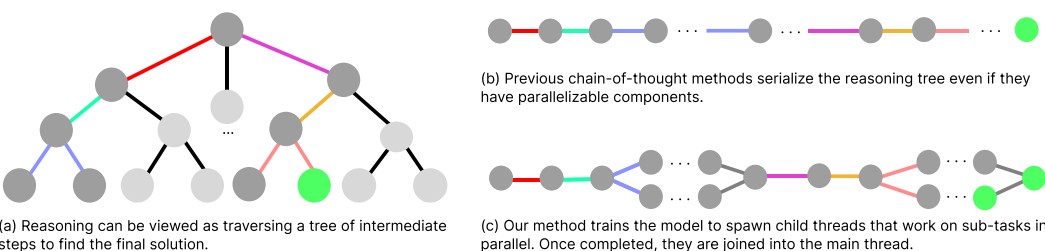

(a) Reasoning can be viewed as traversing a tree of intermediate steps to find the final solution.

(b) Previous chain-of-thought methods serialize the reasoning tree even if they have parallelizable components.

(c) Our method trains the model to spawn child threads that work on sub-tasks in parallel. Once completed, they are joined into the main thread.

Figure 2: **Overview of Adaptive Parallel Reasoning (APR).** While previous chain-of-thought methods directly linearize the reasoning tree, Adaptive Parallel Reasoning alternates between the parent thread and parallel child threads to traverse the reasoning tree more efficiently.

inference-time compute budget. We first describe our task setup, followed by our novel multi-thread setup for parallel search at inference time, and finally the corresponding optimization procedure that combines supervised training and RL fine-tuning.

### 3.1 Preliminaries

**Stream of Search**    Gandhi et al. (2024) propose Stream of Search (SoS), which serializes search traces for reasoning tasks as natural language strings; they train language models to generate SoS strings through supervised training and then further optimize the model through policy improvement algorithms, including STaR (Zelikman et al., 2022) and APA (Zhu et al., 2024). Experiments show this approach allows models to search effectively and to self-improve by adapting and discovering new search and reasoning strategies. However, the serialized reasoning traces of SoS are naturally length-constrained by the context window, and can lead to high latency during inference due to requiring sequential, token-by-token generation. Our approach addresses both limitations by training language models to adaptively parallelize their search at inference time.

**Task: Countdown**    Following existing work that prototypes language model reasoning algorithms (Yao et al., 2023; Gandhi et al., 2024; Pan et al., 2025), we perform experiments on the Countdown task, where a model must map from a set of four numbers to an arithmetic expression that uses each number exactly once and whose value exactly matches a given target (e.g., given $\{1, 4, 6, 8\}$ and target 10, one valid solution is $(8 - 6) \times (4 + 1) = 10$).

### 3.2 Multi-Threading at Inference Time

Inspired by the multi-threading functionality of operating systems, where a process distributes computation concurrently across multiple CPU cores, our framework outlines a generic approach to parallel inference using multi-threaded model execution. At inference time with Adaptive Parallel Reasoning (APR), reasoning proceeds via an adaptively orchestrated hierarchy of parent-child threads. An example trajectory is illustrated in Figure 3.

**Spawning child threads with `spawn()`**    Existing reasoning methods rely on search structure provided by model developers either through prompting (e.g., Press et al., 2023) or through external orchestration and synthesis of inference calls (e.g., Wang et al., 2023). In contrast, in Adaptive Parallel Reasoning, search structure is determined at inference time entirely by the model itself. To support this, we provide the model access to a `spawn(msgs)` operation, which, when selected during decoding, will spawn multiple child threads that are executed in parallel.[1] `spawn()`'s only argument, `msgs`, is a list of strings, each corresponding to a child node to spawn and the context it is passed. Coordination between child threads is achieved by having the parent thread pass each child a distinct context.

**Child thread inference**    Child threads independently yet simultaneously execute inference using the same language model as the parent thread, with each thread's context limited to the tokens passed to it by the parent thread in `msgs`. Similar to other approaches like

---

[1]We use the notation of a "thread" to indicate a consecutive decoding of tokens. The actual implementation depends on the serving framework.

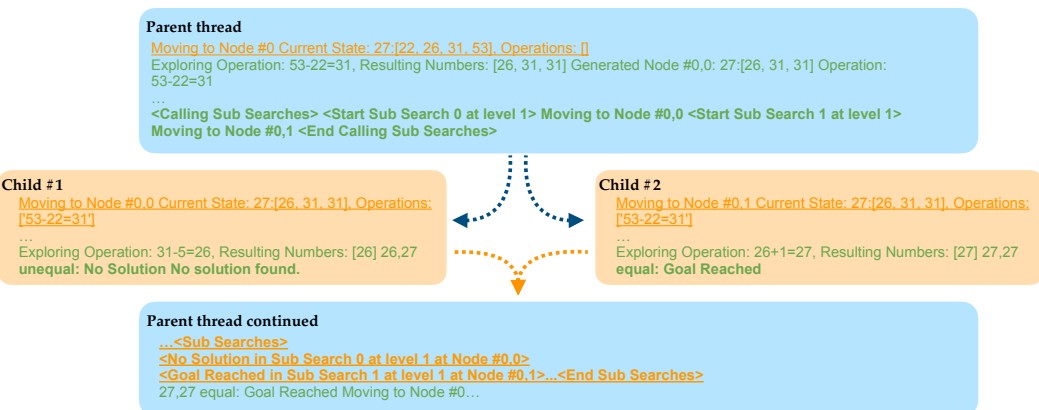

Figure 3: **An example trajectory of our APR method solving the Countdown task.** The model starts with the parent thread, solving the task through reasoning in language, and generates **spawn** commands to instantiate two child threads for parallel reasoning, which are **join**ed back when finished, after which the main thread continues decoding. Orange denotes prefix input tokens, while Green denotes LM-generated tokens. Bold parts correspond to the **spawn** and **join** operations.

self-consistency (Wang et al., 2023), parallelization in our approach affords diversification across reasoning traces. However, because child threads may be conditioned on different contexts provided by the parent thread, they can execute distinct reasoning subtasks and thus avoid the problem of lack of coordination as in independent inference methods like self-consistency.

When a child thread generates the join(msg) operation, it terminates its inference, specifying a sequence of tokens to return to the parent thread as msg. Critically, child threads have control over what they return to the parent thread. In the Countdown task, child threads discard intermediate traces and return only the successful solution path, enabling concise and targeted communication back to the parent. No solution paths are returned for child threads with failed searches.

**Synthesis of child threads after `join()`** Once all child threads have terminated and returned corresponding messages, the parent thread execution continues, conditioned only on its previous context before spawning the child threads and messages returned by the child threads. Keeping intermediate search traces only to the child threads reduces the parent thread's token usage and thus the overall computational cost of inference.

### 3.3 Training Models to Adaptively Parallelize their Reasoning

Enabling language models to effectively use the spawn() and join() operations introduces two major challenges. First, pre-trained language models are simply not trained to call such operations at inference time.[2] Following Gandhi et al. (2024), we use automatically-generated demonstrations to train a reasoning model from scratch with a supervised objective. In our case, we generate demonstrations of reasoning traces that use spawn() and join(). Second, even with supervised learning, the model is trained only to imitate demonstrations, without itself exploring the space of possible search structures which may include structures that are more computationally efficient or effective for the task. To address this, we fine-tune the pretrained model using end-to-end reinforcement learning.

**Supervised initialization of parallel reasoning** We use supervised learning to train a language model from scratch to generate reasoning traces that use spawn() and join(). Following Stream of Search (Gandhi et al., 2024), we use a symbolic solver to generate reasoning traces. In SoS, each demonstration represents a single search strategy – either

---

[2]Following existing work that proposes new learning algorithms for reasoning (Gandhi et al., 2024), we focus on training reasoning models from scratch. We leave experiments on adapting language models pre-trained on general web corpus for future work.

depth first search (DFS) or breadth first search (BFS). To diversify the types of search trajectories and allow for spontaneous parallelization behaviors, we instead develop *hybrid* search that includes examples of both BFS and DFS in the same search trace, which we also empirically find to have slightly better performance.

We implement two symbolic solvers: SoS+, which adapts the original SoS solver to produce serialized hybrid search paths without `spawn()` and `join()`; and APR, which creates hybrid demonstrations with parallelization.[3] For the APR symbolic solver, during its execution, it selects certain nodes to explore in parallel. We delegate the search under these nodes to child LM threads, while the parent thread handles the rest.

Because APR decomposes search into multiple demonstrations, APR faces much less of a context window bottleneck than SoS+ both for training and inference, as each demonstration only includes tokens from part of the search, rather than the full search. Thus, APR can represent much more extensive searches.

**Reinforcement learning for end-to-end policy optimization**   While supervised training establishes a baseline understanding of parallel execution, it merely guides the model to imitate the provided demonstrations, and does not optimize computational efficiency or reasoning effectiveness. We further fine-tune the language model with fully end-to-end reinforcement learning (RL). During RL-based finetuning, we iteratively sample reasoning traces from our current policy, assign them a reward according to the correctness of the proposed solution, and optimize policy parameters with GRPO (Shao et al., 2024). In this stage, the model learns to strategically determine when, how, and how broadly to invoke child threads, maximizing performance by balancing the trade-offs between parallel exploration and the context window constraint.

## 4   Experiments

We evaluate the effectiveness and efficiency of Adaptive Parallel Reasoning compared to serialized chain-of-thought reasoning. We first demonstrate that Adaptive Parallel Reasoning effectively trains models to distribute inference-time computation across multiple inference threads, none of which require the full search context, resulting in a more efficient use of a model's limited context window, and thus supporting scaling to higher compute. We show that under the same latency and context window constraints, Adaptive Parallel Reasoning achieves superior performance. We also find that end-to-end reinforcement learning significantly enhances the effectiveness of Adaptive Parallel Reasoning, and that when tasked to optimize the accuracy end-to-end, RL results in wider, in addition to deeper searches, indicating the effectiveness of adding another dimension of scaling to the method.

**Experiment setup**   Throughout our experiments, we use a standard decoder-only language model trained from scratch with the Llama2 architecture (Team, 2023). The model employs the Llama2 tokenizer, contains 228M non-embedding parameters, and supports a 4,096-token context window. All models are initialized via supervised learning on 500k trajectories generated using Countdown symbolic solvers. For our baseline, we use trajectories from the SoS+ solver, while for our method we use trajectories from the APR symbolic solver.

To directly measure the compute-accuracy trade-off, we adopt a budget constraint method similar to Chen et al. (2021), training length-controlled models by conditioning on context-window sizes for SoS+ models. Specifically, we partition training sample lengths into bins of 512 tokens, using these bin sizes as conditioning signals during training. At inference time, specifying a bin size allows us to control the number of generated tokens. For APR models, however, conditioning on context window size is less suitable as child thread lengths can vary significantly. We condition these models on the number of child threads initiated per parent thread, which strongly correlates with the total number of tokens across all threads.

We use SGLang (Zheng et al., 2024) for inference due to its high-performance and support for continuous batching and radix attention, which enables efficient running of APR.

---

[3]We provide details of both symbolic solvers in Appendix A.6, including pseudocode in Algorithms 1 (SoS+) and 2 (APR).

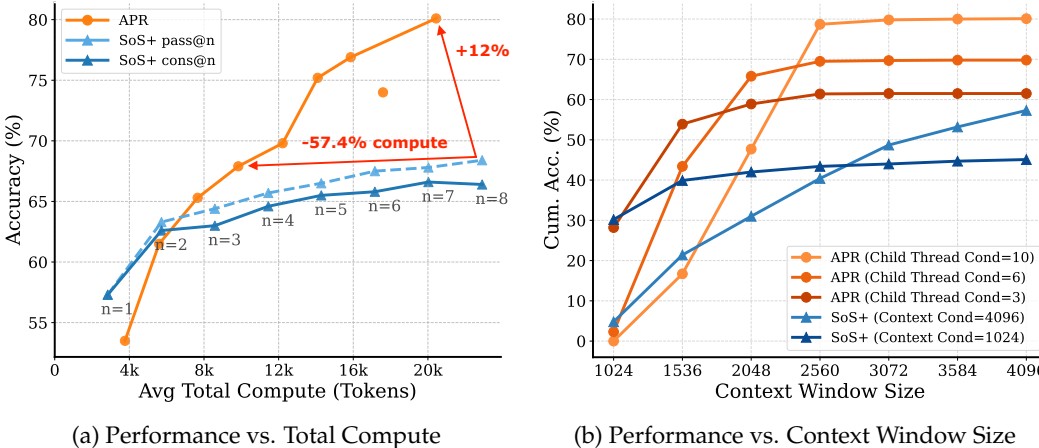

(a) Performance vs. Total Compute          (b) Performance vs. Context Window Size

Figure 4: **Scaling performance of SoS+ vs APR.** (a) APR achieves higher accuracy with increasing compute budget compared to SoS+ with Best-of-N sampling. (b) APR more effectively utilizes fixed context windows across different thread configurations. The cumulative accuracy measures the total accuracy considering only the outputs that fall within the context window constraint—that is, only outputs no longer than the set constraint are counted as correct.

**Baselines**    We compare our method against two baselines: search via long chain-of-thought reasoning (SoS+) and self-consistency selection (denoted **cons@n**), the standard parallel inference method for scaling test-time compute. The cons@n approach is implemented by independently sampling reasoning traces from SoS+, excluding outputs that fail to find a valid solution, then applying majority voting to the final search paths from the remaining outputs. The most frequently occurring solution is then selected as final answer. Additionally, we report **pass@n**, the rate at which at least one returned solution is correct, to illustrate the upper-bound performance achievable through repeated sampling with simple ensemble-based parallel inference.

**Evaluation metrics**    We consider two categories of metrics to evaluate the model performance and efficiency. First, we measure **accuracy** as the percentage of Countdown problems in the test set successfully solved by the model. Second, we evaluate computational efficiency by measuring the **total tokens** generated during reasoning, which reflects the model's compute usage at test time. Additionally, we evaluate **latency** using two complementary metrics: **sequential tokens**, representing the maximum number of causally dependent tokens that must be processed sequentially (i.e., parallel sub-thread generation considers only the longest token sequence among independently processable threads); and **real-world latency**, the wall-clock time required to solve a single task.

## 4.1    Bootstrapping Adaptive Parallel Reasoning from Supervised Training

Following Gandhi et al. (2024), we pre-train our model by imitating sub-optimal search traces generated by the symbolic solver. We construct a training dataset of 500k Countdown problems with corresponding search traces using both SoS+ and APR symbolic solvers. Our results indicate that APR consistently outperforms SoS+ across multiple dimensions.

**Scaling with higher compute**    We first analyze how allocating additional test-time compute improves performance. We measure compute by total tokens generated during reasoning; for SoS+, this is the length of the search trace, while for APR, it is the total token count accumulated across all parent and child calls. We scale test-time compute of APR by varying the number of conditioned child threads from 0 to 10. For SoS+, the context window size is fixed at the maximum number of 4,096 tokens. To scale compute for SoS+ beyond this window, we report cons@n for sample sizes ranging from 1 to 8. We also report pass@n as an upper bound. All scaling experiments use a sampling temperature of 1.0. As shown in Fig. 4a, we find that APR initially under-performs in low-compute regimes (below 4k tokens, or pass@1), which we attribute to "parallelism overhead" - some of the generated

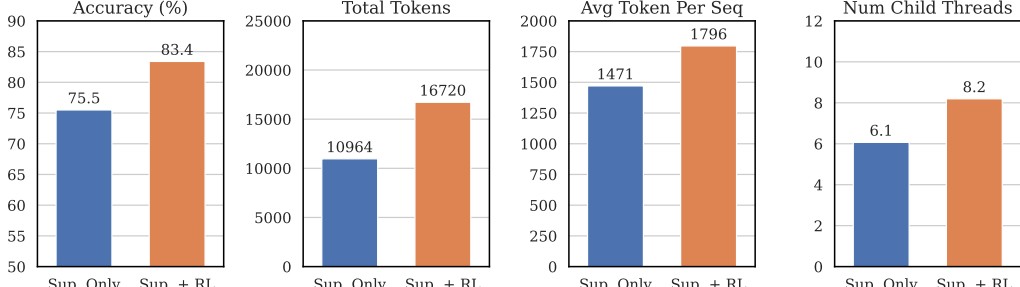

Figure 5: **Comparison of model performance and statistics before and after reinforcement learning (RL).** RL significantly improves performance by optimizing the APR policy beyond what can be learned from supervised imitation learning. Post-RL models strategically increase sequence length (total tokens and avg token per seq) and, more substantially, child thread count (the number of spawned child threads), demonstrating an advantage of broadening rather than deepening search in reasoning. This optimization results in substantially higher accuracy (from 75.5% to 83.4%).

tokens are used to orchestrate threads rather than directly contribute to the search. However, as compute increases, APR significantly outperforms SoS+, achieving a 13.5% absolute improvement (66.6% → 80.1%) over SoS+ cons@7 at 20k tokens. It also surpasses SoS+ pass@8 by 11.7% (68.4% → 80.1%) at 24k tokens. Notably, it matches the pass@8 performance of SoS+ while consuming 57.4% less compute. This clearly indicates that APR scales more effectively with increased compute by enabling parallel exploration via independently executed child threads. Importantly, this parallel design maintains efficiency with minimal impact on latency, as further detailed in §4.3.

**Scaling with context window size** We also evaluate the performance of SoS+ and APR under varying context window constraints (1k to 4k tokens). For each setting we sample once and report cumulative accuracy: at every window size we count only those traces whose length is within that limit. Budgets below 1k are omitted because neither method can produce usable solutions. To investigate performance trade-offs, APR is evaluated with 3, 6, and 10 child threads, whereas SoS+ is trained with context conditioning at 1,024 and 4,096 tokens. The 1,024-conditioned SoS+ occasionally generates traces slightly longer than 1,024 tokens; these same outputs become valid when the limit is relaxed, so its curve remains meaningful beyond the 1,024 window. As illustrated in Fig. 4b, APR consistently exploits context more efficiently. With only 3 child threads, it already surpasses SoS+ at every window size, and with 6 or 10 threads, it achieves around 10% and 20% higher accuracy, respectively, at the 4k-token limit. The advantage comes from distributing reasoning across parallel threads, enabling more total tokens without packing the entire trace into one context window—a limitation inherent to serialized SoS+. Overall, APR more effectively exploits fixed context budgets, yielding significantly better performance.

## 4.2 End-to-end Policy Optimization through Reinforcement Learning

We employ reinforcement learning (RL) to optimize the APR policy end-to-end, with implementation details presented in Appendix A.2. As shown in Fig. 5, RL significantly improves model performance across different initial conditions, resulting in substantially higher accuracy (from 75.5% to 83.4%). Pre- and post-RL models exhibit markedly different behaviors. RL increases both the sequence length (from an average of 1,471 to 1,796 tokens; 22.1% relative increase) and number of child threads (from an average of 6.1 to 8.2 child threads; 34.4% relative increase). This implies that, for the Countdown task, a broader search is more optimal than a deeper one—as discovered by RL.

## 4.3 Efficiency of APR

We demonstrate that APR improves reasoning efficiency both theoretically and practically compared to SoS+ serialized chain-of-thought baselines (by measuring sequential token

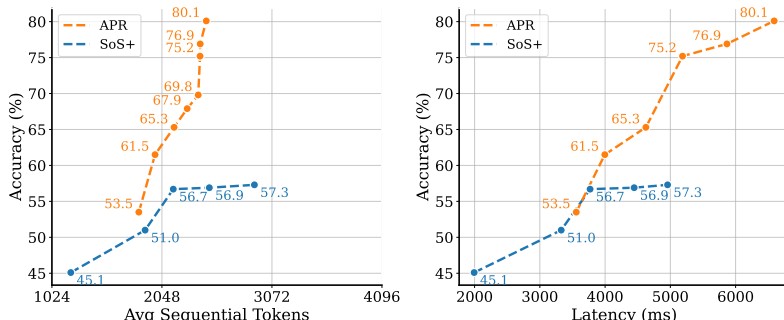

Figure 6: **Efficiency comparison between APR and SoS+.** Left: accuracy vs. sequential token usage. Right: accuracy vs. wall clock latency. APR consistently achieves higher accuracy with fewer tokens and lower latency, demonstrating superior efficiency.

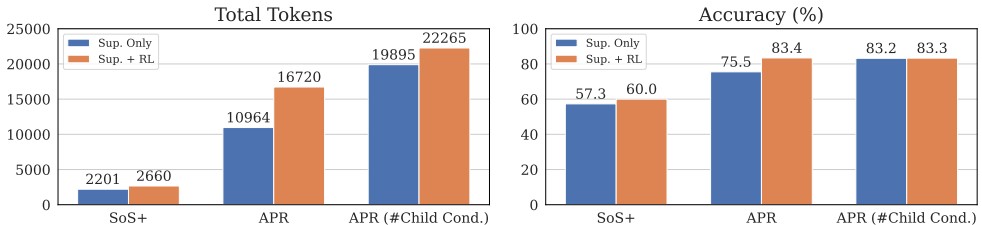

Figure 7: RL provides a larger performance gain to APR (7.9%) compared to SoS+ (2.7%), resulting in substantial advantages of APR over SoS+ after RL (83.4% vs 60.0%). Ablations with the number of child threads enforced show that such performance gain mainly comes from teaching the model how to allocate compute resources instead of making more optimal decisions with the same resources.

usage and real-world latency). We define sequential token usage as the average length of the longest non-parallelizable component across parent and child threads, serving as a lower bound on sequential computation. For SoS+, this directly equals the average output sequence length. Fig. 6 shows accuracy against sequential token usage for both methods. Results confirm that APR effectively leverages parallel exploration, significantly boosting accuracy with minimal additional sequential tokens beyond 2,048 tokens, rarely exceeding 2,500 tokens. In contrast, SoS+ achieves only marginal accuracy improvements despite quickly approaching 3,000 tokens, highlighting the scaling advantages of parallel exploration under constrained contexts.

Moreover, we evaluate the practical efficiency of APR by analyzing real-time latency. Inspired by the spawn() operation commonly used in computer systems, APR is specifically designed to leverage hardware parallelization effectively. We consider an API-based serving scenario where both SoS+ and the main call of APR initially consume the same fixed amount of compute, while subsequent child threads utilize additional computational resources concurrently. Specifically, we deploy the models on an 8-GPU NVIDIA RTX A6000 server, dedicating one GPU to handle the main inference thread, with the remaining GPUs allocated for executing child threads in parallel. We measure the latency per sample across the entire test set. Results in Fig. 6 demonstrate that APR achieves a substantially better accuracy-latency trade-off compared to SoS+. Notably, at a latency of 5000ms per sample, APR reaches an accuracy of 75%, an 18% absolute improvement over SoS+'s 57%. Further performance optimizations are possible and can be found in Appendix A.10, implying the potential of APR for superior performance and efficiency in realistic deployment scenarios.

## 4.4 Ablation Study

**RL on SoS+ vs APR**   As shown in Fig. 7, we observe that RL significantly improves the policy in APR, with a significant increase in the sequence length and the number of child

threads the model uses. To evaluate RL's impact on our baseline, we also applied GRPO to SoS+. As expected, RL increased the sequence length and improved SoS+ performance as well. However, the accuracy improvement for SoS+ (2.7%) was substantially smaller than for APR (7.9%). This difference in improvement rates correlates with changes in total token usage. While SOS+ only increased total tokens by 20.9%, APR showed a 52.5% increase. These results demonstrate APR's effectiveness in allowing RL to scale token usage beyond context window limitations by distributing computation across multiple child threads.

**Disentangling contributing factors in RL** As shown in Fig. 7, RL significantly improves policy accuracy. However, this improvement may stem from two distinct factors: (1) test-time scaling, where RL encourages the model to use more tokens at test time and perform more extensive search (i.e., try more options); or (2) reasoning-quality improvements, where RL teaches the model to search more effectively (i.e., select options more cleverly).

To disentangle these factors, we conducted an experiment where we conditioned the model to use the maximum number of child threads (10 threads) both before and after RL training, so that it almost always uses the most test time compute possible even before RL. Our results reveal that (1) with maximum child threads enforced, we observe minimal changes in both thread count and sequence length after RL; and (2) accuracy remains nearly identical pre- and post-RL under these fixed compute conditions, from 83.2 to 83.3% accuracy. This indicates that in our experiments, the primary benefit of reinforcement learning comes from scaling test-time compute rather than improving decision quality within a fixed budget.

# 5 Conclusions, Limitations, and Future Work

We presented Adaptive Parallel Reasoning, which enables language models to adaptively distribute computation across serial and parallel reasoning paths using a parent-child threading mechanism. In this method, we employed supervised training and reinforcement learning to enable models to learn to develop parallelization strategies without manually designed structures. Supervised training establishes a baseline understanding of parallel execution, and to further optimize the effectiveness of reasoning, we fine-tune the language model with fully end-to-end reinforcement learning.

Experiments on the Countdown task demonstrate that APR achieves: (1) higher performance within the same context window (83.4% vs. 60.0% at a 4k context); and (2) superior scaling behavior as compute budgets increase, resulting in more substantial performance improvements (80.1% vs. 66.6% using 20k tokens); (3) significantly higher success rates compared to serialized search methods at equivalent latency constraints (75.2% vs. 57.3% at around 5,000 ms). Overall, APR demonstrates the potential of reasoning systems that dynamically structure their inference processes to achieve improved scalability and efficiency.

We see many research directions that we are excited to explore in the future:

**1. Extending to pre-trained language models and general tasks.** Currently, our experiments are restricted to non-pretrained LMs on Countdown tasks. Moving forward, we plan to extend our methods to general reasoning tasks with pre-trained LMs, which would validate the approach's broader applicability.

**2. Reducing reliance on supervised training.** Our current setup requires bootstrapping by mimicking the symbolic solver. We aim to use strong pre-trained LMs as initial checkpoints and, motivated by DeepSeek R1-Zero (DeepSeek-AI, 2025), explore whether we can bypass the supervised training stage and directly apply reinforcement learning.

**3. Better orchestration and communication protocols.** As a first step, this work uses `fork` and `join` operations for inter-thread communication and thread orchestration. We plan to explore other protocols, such as any-to-any messaging, all-to-all communication patterns, or subscription-based methods.

**Acknowledgements**   The authors would like to thank Nicholas Tomlin, Boyi Li, Li Erran Li, Sanjay Subramanian, Ruiqi Zhong for their valuable discussions. This work was partially supported by compute resources provided through Google's TPU Research Cloud (TRC) and Nvidia, and an Ai2 Young Investigator Award.

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

# A Appendix

## A.1 Additional Related Works

**Language model reasoning algorithms** Chain-of-thought (CoT) prompting (Wei et al., 2022) was the first approach to show how, with the right prompt, additional test-time compute can be exploited to improve language model performance on an end task. Building on the promise of CoT, models were evaluated on increasingly complex reasoning tasks, and the output sequences produced by CoT and its variants became increasingly out-of-distribution from model training data. This motivated the development of approaches that fine-tune language models to perform reasoning more effectively by making their reasoning traces in-domain, and by searching for optimal reasoning strategies at training time (Zelikman et al., 2022; Zhu et al., 2024). In these approaches, during learning, reasoning traces are sampled from a language model on training task instances, and the sampled traces are used during model optimization, for example by fine-tuning only on traces that lead to task success, or by reinforcement learning. Such exploration-based learning is the key element of success for popular models like DeepSeek-R1 (DeepSeek-AI, 2025), OpenAI's o1 (OpenAI, 2024), Kimi 1.5 (Team, 2025), and Stream-of-Search (SoS; Gandhi et al., 2024). However, these approaches all result in models that produce significantly longer output sequences, which imposes several limitations due to the fact that autoregressive generation is inherently serial: longer output sequences means higher latency, and context window constraints on serving system cap the feasible output length and complexity of reasoning. We also experiment with exploration-based learning for reasoning, but with a crucial difference from these prior approaches: we provide a special action to the language model, allowing it to distribute their inference-time computation into serial *and* parallel operations. In comparison to serialized inference, parallelizing reasoning traces both reduces latency and the stress of context window limitations. Not only do our experimental results confirm this, but we also show that models trained end-to-end to distribute their inference-time compute significantly outperform these existing approaches at the same computational budgets.

Parallelizing inference with multiple independent language model calls has emerged as another avenue for scaling reasoning tasks. For example, in self-consistency (Wang et al., 2023), multiple candidate solutions to a reasoning problem are generated independently, and the most common solution among them is chosen. While ensembling methods like these effectively improve task performance, their core limitation is the independence between inference threads; the lack of coordination among threads leads to redundant computation and thus suboptimal resource utilization.

Methods such as Tree-of-Thought (Yao et al., 2023), Graph-of-Thought (Besta et al., 2024), Skeleton-of-Thought (Ning et al., 2024), Atom-of-Thought (Teng et al., 2025), and Self-Ask (Press et al., 2023) build on these simple ensembling methods by structuring the exploration of reasoning paths into multiple parallelizable calls to run inference on a language model. Several recent approaches also propose to perform reasoning tasks by simulating multi-agent interaction (Du et al., 2023; Sel et al., 2024; Kim et al., 2024; Zhang et al., 2024; Zhuge et al., 2024; Hooper et al., 2025; Wang et al., 2025). However, the proposed search structures (and, in the case of multi-agent reasoning, the communication protocol) of these methods are fixed and hand-designed, which limits their flexibility and scalability. Furthermore, these structured inference methods are predominantly prompting-based,[4] without end-to-end optimization, which imposes a wide distributional gap between sequences processed at training and inference-time. Instead, we train language models to structure their own search at inference time, distributing computation between parallelized and serialized operations. In theory, our framework could result in language models that implement the same search structures as existing approaches, such as Tree-of-Thought (Yao et al., 2023), without explicit prompting or hand-designed orchestration of language model calls.

Concurrent to our work, PASTA (Jin et al., 2025) and Hogwild! Inference (Rodionov et al., 2025), released in February and April 2025 respectively, also explore parallel reasoning.

---

[4]While Zhuge et al. (2024) employ reinforcement learning for optimizing prompts and connectivity between agents, the underlying language models are still fixed.

PASTA decomposes a sequential task into parallel sub-tasks that eventually merge back into a single main thread. Our method, however, spawns multiple exploratory threads in parallel and selectively incorporates only successful outcomes, effectively scaling to better performance as compute increases. Hogwild! Inference employs parallel workers to collaboratively solve problems; yet, it remains a prompting method, while APR optimizes collaborative reasoning in an end-to-end manner through supervised training and reinforcement learning, providing a more integrated solution to efficient inference parallelization.

Our work is related to existing algorithms for training language models to directly control the inference process, for both efficiency and performance reasons. For example, PENCIL (Yang et al., 2025) trains models to adaptively discard parts of their context at inference time to reduce the context length, and thus the time complexity of producing a successful reasoning trace. Goyal et al. (2024) train language models to inject *additional* "pause" tokens into their output sequences, which are shown to improve end-task performance by increasing inference-time computation. Both approaches train using supervised demonstrations of token discarding (or injection). To the best of our knowledge, we are the first to use end-to-end reinforcement learning to optimize language model control over inference.

**Language model serving systems** Language model inference is primarily memory-bounded, with modern serving frameworks optimizing performance through batched processing of multiple requests in parallel. Systems like vLLM (Kwon et al., 2023) and SGLang (Zheng et al., 2024) have significantly improved token throughput per request and total throughput across all requests. While these systems can scale horizontally by adding more GPUs or increase total throughput by expanding batch size, they face a fundamental constraint: the token generation rate per individual thread remains limited (typically 50-100 tokens per second for models like GPT-4o). This limitation creates significant latency issues for sequential chain-of-thought methods, as generating reasoning traces for complex problems can take 10-60+ seconds, degrading user experience. Additionally, long reasoning traces quickly consume the available context window. Adaptive Parallel Reasoning addresses these bottlenecks by distributing computation across parallel sub-threads, reducing end-to-end latency while enabling more complex reasoning within practical time constraints and maintaining compatibility with existing serving systems.

## A.2 Implementation Details

**Model architecture** We use a model following Llama2 (Team, 2023) with 228M non-embedding parameters (293M total parameters) with the following key specifications: 18 hidden layers, 1024 hidden dimension, 16 attention heads, and a 4096 token context window.

**Supervised training** We conduct supervised training using 128 TPUv3 cores. Across all experiments, we use a batch size of 256 and a learning rate of 5e-5. The model is trained for 19,000 steps—approximately 10 epochs—following the setup in Gandhi et al. (2024).

**Reinforcement learning** We run reinforcement learning with 2 Nvidia GPUs with GRPO algorithm (Shao et al., 2024). The training batch size is 64, with each sample rolled out 5 times (i.e., the group size is 5). The model is rolled out with SGLang. The temperature for rollout in training is set to 1.0. We use greedy sampling for evaluation unless otherwise stated, since it achieves the best performance for both SoS+ and APR compared to other evaluation temperatures.

Training is conducted using a learning rate of $1 \times 10^{-5}$ and a PPO clip ratio of 0.2. We run training for 150 total steps, each consisting of 2 inner PPO optimization steps. We apply gradient clipping with a maximum norm of 1.0 and validate performance every 25 steps. For the KL divergence factor used in the GRPO framework, we set the factor to 0.01 for SoS+ baseline and 0.001 for APR for training stability.

## A.3 Additional Results with Larger Models

We evaluated both APR and the SoS+ baseline using a 600M-parameter Llama2 model to study scaling with model size. Figure 8 reports accuracy over average total compute (in

thousands of tokens), following §4.1. Both methods improve with increased model capacity, and APR maintains a substantial lead over SoS+ at every compute level.

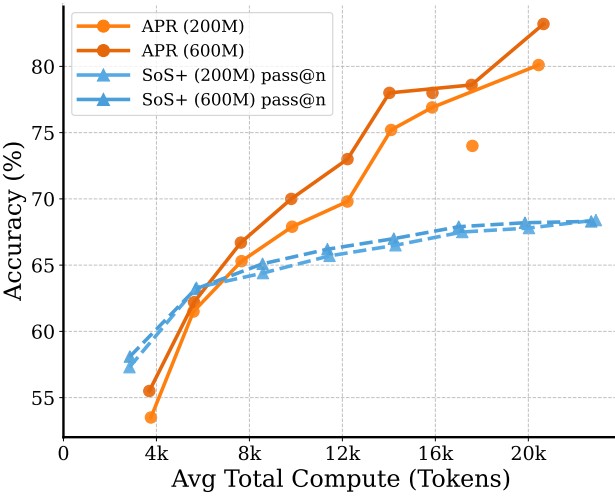

Figure 8: Accuracy vs. average total compute (in thousands of tokens) for 200M and 600M parameter models. APR consistently outperforms SoS+ and scales more strongly with model size.

## A.4 Additional Results with Pretrained Models

To verify that APR's gains extend beyond models trained from scratch, we fine-tuned a pretrained Qwen2.5 1.5 B model on the same SoS+ and APR demonstration data (approximately 4 k supervised steps). As Table 1 shows, APR significantly outperforms the SoS+ baseline on Qwen, mirroring the trend observed with Llama2. This demonstrates that our parallel reasoning framework is agnostic to model family, size, and pretraining status.

|        | Llama2 200M | Qwen2.5 1.5B (pretrained) |
|--------|-------------|---------------------------|
| SoS+   | 57.4 %      | 57.5 %                    |
| APR    | **83.2 %**  | **80.2 %**                |

Table 1: Accuracy of SoS+ and APR when fine-tuning on a pretrained Qwen2.5 1.5 B model versus a Llama 2 200 M model.

## A.5 Extended Context Window Experiments

We evaluated APR and SoS+ on a five-number Countdown variant—whose search space is 40× larger—using context budgets up to 8k tokens. Figure 9 shows that APR continues to improve up to about 6k tokens and outperforms SoS+ beyond 3.5k tokens, achieving gains of 7% and 11% at an 8k-token budget.

## A.6 Symbolic Search Algorithm

We build upon the BFS and DFS algorithm in Stream-of-Search (Gandhi et al., 2024) and implement our improved version SoS+ as shown in Algorithm 1.

For the parallel search version, we mostly follow the same algorithm, but when the model decides to do DFS on a certain node, it will spawn a list of sub-searches and explore the nodes in parallel. The algorithm is illustrated in Algorithm 2. We recommend referring to our official implementation available at github.com/Parallel-Reasoning/APR.

---

**Algorithm 1** SoS+ Symbolic Solver

---

**Require:** start_state, goal_state
 1: state_deque ← SE(start_state)
 2: **while** state_deque ≠ ∅ **do**
 3:   current_state ← state_deque.popleft()
 4:   **if** current_state = goal_state **then**
 5:     **return** "Goal reached"
 6:   **end if**
 7:   **if** IS_PROMISING(current_state) **then**
 8:     result ← SOS+(current_state, goal_state)
 9:     **if** result = "Goal reached" **then**
10:       **return** "Goal reached"
11:     **end if**
12:   **else**
13:     states_to_explore ← SE(current_state)
14:     state_deque.extend(states_to_explore)
15:   **end if**
16: **end while**
17: **return** "No result"

---

**Algorithm 2** APR Symbolic Solver

---

 1: **function** APR(start_state, goal_state, main = True)
 2:   state_deque ← SE(start_state)
 3:   **while** state_deque ≠ ∅ **do**
 4:     current_state ← state_deque.popleft()
 5:     **if** current_state = goal_state **then**
 6:       **return** "Goal reached"
 7:     **end if**
 8:     states_to_explore ← SE(current_state)
 9:     **if** main ∧ IS_PROMISING(current_state) **then** ▷ Parallel exploration in promising states
10:       parallel_results ← [ APR($s$, goal_state, False) | $s$ ∈ states_to_explore]
11:       **for** r ∈ parallel_results **do**
12:         **if** r = "Goal reached" **then**
13:           **return** "Goal reached"
14:         **end if**
15:       **end for**
16:     **else**
17:       state_deque.extend(states_to_explore)
18:     **end if**
19:   **end while**
20:   **return** "No result"
21: **end function**

---

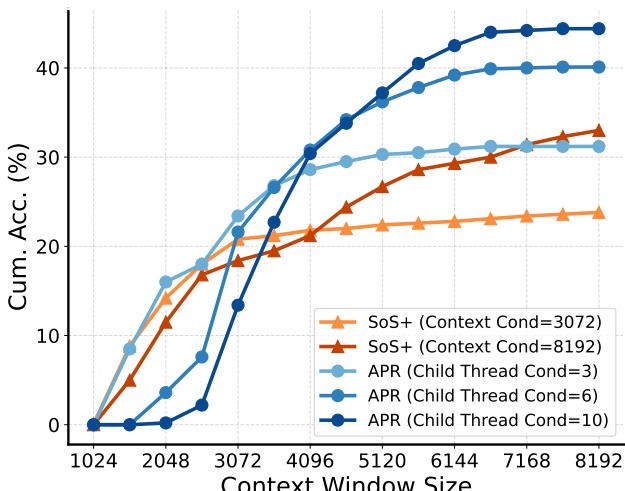

Figure 9: Accuracy vs. context window size on the five-number Countdown task for SoS+ and APR.

SE is a state expansion function that decides what are the next step to explore, while is_promising a heuristic function to judge if the current node is promising or not.

Concretely, the state expansion function follows Gandhi et al. (2024), where we select the top K operations ranked by the multiple heuristic. The multiply heuristic considers the target number $T$ and the input numbers and the factors of the target. If $T$ has factors $\{f_1, f_2, ..., f_m\}$, the multiply heuristic can be written as $h_{multiply} = \min(\{|f_j - \sum_{i=1}^{|I|} n_i| \forall j \in [1, m]\})$. Due to the difficulty in implementing an accurate is_promising function, we simply let it return True for 10% of the time, which we leave reinforcement learning to further optimize the promising node selection strategies.

## A.7 SoS vs. SoS+

| Method | Temp = 0.0 | Temp = 0.1 | Temp = 0.5 | Temp = 1.0 |
|--------|-----------|-----------|-----------|-----------|
| SoS    | 49.5%     | 49.6%     | 47.1%     | 37.9%     |
| SoS+   | **57.3%** | **57.1%** | **52.0%** | **48.1%** |

Table 2: Accuracy comparison between SoS and SoS+ across different temperature settings. SoS+ consistently achieves higher performance, especially at lower temperatures.

Here, we compare the performance of models trained with SoS and SoS+ demonstrations without conditions under different sampling temperatures. As shown in Tab. 2, our SoS+ approach significantly outperforms the original SoS method.

## A.8 Effect of SoS+ Supervised Training Data

We study whether improving the quality of supervised demonstrations can narrow the performance gap between SoS+ and APR. Specifically, we evaluate two enhanced training strategies for SoS+: (1) increasing the maximum beam size from 5 to 15 during demonstration generation to mirror the setup used in APR, and (2) applying rejection sampling to curate 500K high-quality training examples that are both correct and fit within the context window. We report results at two decoding temperatures: 0.0 and 1.0.

As shown in Tab. 3, SoS+ performance is fundamentally limited by the context window size, as it cannot learn from long search paths without parallel exploration. Increasing

|                              | Temp = 0.0 | Temp = 1.0 |
|------------------------------|------------|------------|
| SoS+ (baseline)              | 57.3%      | 48.1%      |
| SoS+ (beam size = 15)        | 50.9%      | 42.5%      |
| SoS+ (rejection sampling)    | 56.5%      | 54.5%      |

Table 3: Accuracy of SoS+ trained with different supervision strategies.

| Method | Temp = 0.0 | Temp = 0.1 | Temp = 0.5 | Temp = 1.0 |
|--------|------------|------------|------------|------------|
| *SoS+* | 57.3% | 57.1% | 52.0% | 48.1% |
| + RL | 60.0% | 59.3% | 59.1% | 58.1% |
| *APR* | 75.5% | 75.9% | 74.9% | 67.8% |
| + RL | 83.4% | 82.9% | 81.3% | 76.4% |
| *APR (Num Child Thread Enforced)* | 83.2% | 83.3% | 83.8% | 80.1% |
| + RL | 83.3% | 83.3% | 83.8% | 81.9% |

Table 4: Accuracy of different search strategies across sampling temperatures. Each base method is followed by its variant trained with RL.

the beam size improves symbolic solver accuracy but leads to longer trajectories that exceed the model's context length, ultimately reducing performance. Rejection sampling improves training quality by filtering for context-bounded correct examples, yielding gains at temperature 1.0, but the overall impact remains modest and still falls short of closing the gap to APR.

### A.9 Effect of Temperature

We ablate the effect of temperature on how performance scales with compute, extending Fig. 4a. As shown in Fig. 10, APR consistently outperforms SoS+ regardless of temperature, exhibiting greater robustness and efficiency under varied sampling conditions. Again, Table 4 shows that our results—and their relative advantages—remain consistent across temperatures, both before and after RL.

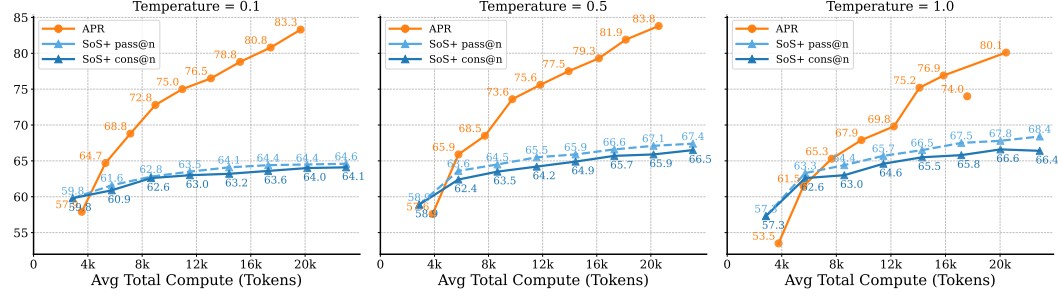

Figure 10: **Ablation of APR and SoS+ vs. total compute under different temperature settings.** Across all settings, APR demonstrates more reliable and efficient scaling behavior.

### A.10 Differences Between Sequential Tokens and Wall-Clock Time

While the number of sequential tokens correlates well with wall-clock time, in practice, there is a slight mismatch due to hardware constraints. Specifically, we utilize an 8-GPU server; however, in scenarios with higher compute demands—such as those involving up to 10 child threads—some GPUs may end up handling multiple child threads, leading to uneven workloads and increased computational load on certain devices. This issue can be mitigated by allocating additional GPUs and improving load balancing during model serving.

