# OpenReview forum: "Learning Adaptive Parallel Reasoning with Language Models"
_colmweb.org/COLM/2025/Conference — COLM 2025_

### Official Review · Reviewer_VxmG · 2025-04-15

**Rating:** 7
**Confidence:** 3
**Ethics Flag:** 1

**Summary:**

This paper introduces Adaptive Parallel Search (APS), a framework allowing language models (LMs) to dynamically decide between serial and parallel computation during reasoning. APS aims to overcome the latency/context limits of serial methods and the inefficiency of uncoordinated parallel approaches. It equips LMs with `spawn()` and `join()` operations, enabling a parent thread to delegate sub-tasks to parallel child threads. Critically, the model *learns* this parallelization strategy end-to-end via reinforcement learning (RL), optimizing task success (Countdown number puzzles) without fixed, predefined structures. The process starts with supervised training on demonstration trajectories containing `spawn/join`, followed by RL fine-tuning. Experiments show APS significantly outperforms serialized search (SoS+) and simple parallelism (cons@n) by achieving higher accuracy at the same latency, better utilizing fixed context windows, and scaling more effectively with compute. RL teaches the model to favor parallel exploration, demonstrating a promising step towards LMs that autonomously optimize their own reasoning processes for efficiency and scalability.

**Questions To Authors:**

1. Could you elaborate on the comparison with Tree-of-Thought (ToT)? ToT also explores parallel branches. Is the key difference that APS learns when to branch and uses RL, whereas ToT typically relies on fixed prompting/structure for branching and evaluation?

2. How crucial is the "hybrid search" (SoS+) symbolic solver for generating the initial demonstrations compared to using pure BFS or DFS as in the original SoS paper? Did you find the hybrid approach necessary for the model to learn effective parallelization later?

3. Regarding adapting this to pre-trained models: What are the main anticipated challenges? Would it involve teaching the model the spawn/join tokens via fine-tuning, and how might that interact with its existing capabilities?

4. The RL optimization clearly prefers increasing child thread counts. Does this suggest that for this task/model size, parallel exploration is almost always better than deeper serial exploration once a basic level is reached? Do you foresee scenarios or tasks where the learned policy might favor serial generation more heavily?

5. How does APS handle failures in child threads? Fig 1 shows "no solution found." Does the parent thread gain any information from failed branches, or are they simply discarded upon join()?

6. The latency measurement setup uses 1 GPU for the parent and others for children. How sensitive are the latency results (Fig 5b) to the number of GPUs available for child threads? What happens if the number of spawned threads exceeds available dedicated GPUs?

**Reasons To Accept:**

1. Novel Synthesis for Adaptive Parallelism: The core contribution – training an LM via RL to learn when to dynamically switch between serial and parallel execution using spawn/join primitives – appears novel. It elegantly combines concepts from parallel search, learned policies, and LM generation, moving beyond fixed structures (like ToT) or purely serial learned search (like SoS+RL).

2. Strong Empirical Performance: APS demonstrates significant advantages over strong baselines (SoS+, cons@n) on the Countdown task across multiple crucial dimensions: accuracy at fixed latency, accuracy within fixed context limits, and scaling with compute budget. The ~18% absolute accuracy improvement at 5000ms latency is particularly compelling.

3. Addresses Key LLM Limitations: The method directly tackles the high latency and context window constraints of sequential reasoning, and the inefficiency/lack of coordination in naive parallel approaches, offering a principled way to balance exploration breadth and depth.

4. End-to-End Learned Control: Training the model itself to control the parallelization strategy via RL (optimizing task success) is more flexible and potentially more powerful than relying on fixed, hand-designed prompts or external orchestrators used in many prior structured reasoning works. The RL step demonstrably improves performance over just supervised imitation.

5. Clear Methodology and Evaluation: The paper clearly outlines the APS mechanism, the training process (supervised + RL), and the experimental evaluation, including relevant metrics like sequential tokens and wall-clock latency.

**Reasons To Reject:**

1. Limited Task Domain: The experiments are solely focused on the Countdown task. While complex and suitable for evaluating search, its synthetic nature and specific structure (number manipulation) might not fully represent the diversity of reasoning tasks LMs face (e.g., open-ended QA, logical deduction, planning). Demonstrating effectiveness on broader reasoning benchmarks would significantly strengthen the claims.

2. Training from Scratch: The models are trained from scratch (Llama 2 228M). While necessary to integrate the spawn/join mechanics and RL, it raises questions about applicability to large, pre-existing foundation models. Adapting APS to fine-tune existing large models might be challenging and is left for future work (Footnote 3). The benefits might differ substantially with much larger base models.

3. Symbolic Solver Reliance: The supervised initialization relies heavily on trajectories generated by a symbolic solver specific to Countdown. This raises concerns about scalability to tasks where optimal/good search paths are not easily generated programmatically. How would APS be bootstrapped for more open-ended reasoning tasks?

4. Complexity and Overheads: Implementing the parent-child threading mechanism, managing contexts for sub-threads, and handling the join operation introduces non-trivial complexity into the inference process. While latency results look promising (Fig 5b), the practical overheads, especially synchronization costs and potential GPU under-utilization if child threads are very short or uneven, need careful consideration in real-world deployment (briefly touched upon in Appendix A.5).

---

> ### Author Response · Authors · 2025-06-03
> **Response**
>
> Thank you for your valuable feedback. Here are our responses to the comments:
>
> **Task selection and model capacity.** In our experiments, we intentionally follow the methodological choices of the Stream of Search (SoS) study (Gandhi et al., 2024; COLM 2024 oral presentation), which centers on the Countdown task due to its structured reasoning challenge, high branching factor, and ease of result verification. Similarly, we train our model from scratch to allow for controlled investigations. As in SoS, focusing on a synthetic yet challenging domain enables rigorous evaluation across accuracy, latency, and scalability, providing clear insights into the effectiveness of our proposed approach.
>
> For model scale, we trained a version that is approximately 3x larger, and the performance trend remains consistent (results shown in our response to Reviewer RZt7), demonstrating that our test-time scaling method is orthogonal to training-time scaling. The size of our pre-trained model (~600M parameters) roughly matches some of the pre-trained models (e.g., Qwen-2.5-0.5B). The fact that APS still significantly outperforms the baseline at this scale confirms that our method remains effective even when applied to models of pretraining-level size.
>
> Regarding generalization, the main innovation of our method (adaptive task decomposition and parallelization optimized through reinforcement learning) is inherently suitable for more domain-general structured reasoning tasks beyond Countdown, e.g., for math reasoning. In this current work, we follow the precedent established by the SoS study of focusing on the well-defined but challenging reasoning task of Countdown. While it is not feasible to include results on benchmarks like GSM8K and HumanEval within the rebuttal period, we plan to explore applying APS to these domains as a next step. We will revise the discussion section to better highlight the generalizability of SoS and outline future directions for extending it to a wider range of tasks.
>
> **Symbolic solver reliance.** While our experiments with APS use symbolic solvers for generating supervision data for pre-training from scratch, our core method is generalizable to broader reasoning domains. For example, we could use LLMs to decompose existing open-ended reasoning tasks (e.g. math word problems) into subtasks, and identify which of the subtasks can be parallelized; these heuristically-generated decompositions could serve as weak supervision to bootstrap APS training with reinforcement learning.
>
> To clarify a misunderstanding, APS does *not* require an optimal search path in its pre-training data from symbolic solvers. Our symbolic solver only needs to create viable search paths (i.e., ones that find *a* solution, not necessarily optimally). The generated demonstrations we pre-train on are suboptimal. We show that we can optimize search paths via end-to-end fine-tuning with reinforcement learning, without requiring an optimal solver.
>
> **Complexity and overheads.** We appreciate the reviewer raising these important considerations for real-world deployment. Careful system design, including optimized scheduling, efficient KV-cache management, and techniques like block-wise computation (akin to frameworks like SGLang), can significantly mitigate these. For instance, very short child threads could be executed on the main thread, and while uneven threads might wait for the longest to complete, freed resources from shorter, completed threads can be dynamically reallocated. While these are important engineering considerations, our work is a preliminary exploration focused on the novel adaptive parallel reasoning capability. The significant performance and efficiency improvements, along with better leverage of parallel hardware, highlight the strong potential that we believe outweighs these manageable challenges for future deployment-focused work.
>
> **Comparing with Tree-of-Thought (ToT).** We thank the reviewer for raising this important comparison. One key difference, as mentioned by the reviewer, is that ToT relies on a fixed prompt / inference structure to determine branching factors. Our proposed approach contrasts with ToT and many other structured inference methods (e.g., multi-agent debate (https://arxiv.org/abs/2305.14325), graph-of-thoughts (https://arxiv.org/abs/2308.09687), etc.) that require manually defining fixed prompt and inference structures. Instead, APS allows a model to adaptively decide at inference time when and how to branch its search.
> Another key difference with ToT (and other fixed-structure inference methods) is that such methods cannot be optimized for search beyond their manually-defined inference structures; they are all based on zero-shot prompting with no model training. Instead, APS initially uses supervised training to bootstrap branching behavior, and then applies reinforcement learning to perform end-to-end optimization of search strategies.

---

> > ### Author Response · Authors · 2025-06-03
> > **Response (Cont.)**
> >
> > **Hybrid search (SoS+).** If we were, like SoS, to pre-train our method on demonstrations that are either entirely breadth-first search (BFS) or depth-first search (DFS), our models would struggle to learn that search can be structured with both BFS and DFS in the same inference run. Thus, using our SoS+ hybrid search is critical for pre-training models that can produce inference structures that adaptively trade off between parallelization and serialization. Table 1 in Appendix A.4 demonstrates that SoS+ also results in stronger pretrained reasoning models when compared to SoS, so SoS+ is indeed a valid baseline for APS.
> >
> > **Adapting to pre-trained models.** Thank you for the thoughtful question. There are two main challenges in adapting APS to pre-trained models to a more general domain:
> > 1. Task Decomposability: Not all reasoning tasks are naturally parallelizable. Tasks with strong sequential dependencies may not benefit from thread forking. In this case, proper engineering in data curation and model training needs to be done to preserve serial reasoning behaviors if not parallelized decomposition is viable.
> > 2. Capability Preservation: Teaching the model spawn/join tokens via fine-tuning is indeed a potential concern. Prior work shows that specialized fine-tuning (e.g., for math or coding) often degrades general abilities like open-ended dialogue. This can be addressed by designing careful training strategies to preserve broad capabilities.
> >
> > **Parallel exploration vs serial exploration in RL.** Currently, our RL formulation does not impose an explicit penalty on the number of threads, so the policy’s preference between parallel and serial exploration depends purely on performance outcomes (our reward function is based entirely on reasoning accuracy, not efficiency). In our experiments, we observe that RL scales both parallel and serial dimensions to maximize the explored search space, with a notable preference toward parallel scaling. This is shown in Fig. 6: For APS, RL scales the number of total tokens from 10k to 19k, achieving 7% performance boost. In contrast for our baseline, RL only scales the number of tokens from 2.2k to 2.7k, achieving 2.5% performance boost.
> >
> > That said, we do foresee scenarios where a learned policy may favor serial reasoning. Tasks with minimal ambiguity, strong step-wise dependency, or low decomposability—such as straightforward math problems—may not benefit much from parallel exploration. In these cases, an RL-trained APS model may naturally adopt a more serial strategy. Even in such settings, we expect APS to perform comparably to non-APS baselines at worst. Moreover, we see potential for APS to outperform in certain serial settings by leveraging subtask decomposition. If a task can be broken into serially executed subtasks, each with a constrained context window, APS can achieve both efficiency (through reduced per-thread context length) and potentially better generalization (by isolating subproblems that are easier to learn). We plan to explore these directions in future work.
> >
> > **Handling failures in child threads.** APS handles failures in child threads by treating unsuccessful explorations as informative signals to avoid further redundant investment in those unpromising directions. When a child thread returns “no solution found,” the parent thread implicitly learns that the corresponding path is unlikely to yield a valid solution. Consequently, the parent thread can effectively redirect resources and exploration efforts toward alternative reasoning paths. For instance, if the first child thread exploring the path “53 - 22 = 31” returns a failure, the model is guided to prioritize other operations or strategies. Thus, these failed subthreads are not simply discarded upon join(); they guide future search. Crucially, since the content is not merged back, they also reduce context usage and improve efficiency by avoiding unnecessary computation and storage.
> >
> > **Latency vs. number of GPUs and number of child threads.** We refer the reviewer to the general response for our latency measurements with varying GPU counts. In summary, when the number of spawned threads exceeds available GPUs, latency increases due to resource sharing. However, our inference engine will batch the decoding process, which softens the increase. As a result, the system maintains reasonable performance, even when threads outnumber GPUs.

---

> > > ### Comment · Reviewer_VxmG · 2025-06-03
> > > **Raising my score to Accept**
> > >
> > > Thank you for conducting the additional experiments on larger models and latency when the number of GPUs change --- I have raised my score to 7.

---

### Official Review · Reviewer_RZt7 · 2025-05-13

**Rating:** 9
**Confidence:** 4
**Ethics Flag:** 1

**Summary:**

* The paper presents Adaptive Parallel Search (APS), a novel framework that dynamically allocates inference-time compute between serial and parallel reasoning via learned spawn()/join() operations.

* The methodology is original in training language models end-to-end with reinforcement learning to autonomously structure their own inference threads.

* Experimental results on the Countdown task show significant gains in accuracy under matched latency, context, and compute budgets compared to both serialized and simple parallel baselines.

* The paper is very well written, with clear motivation, a solid and thorouhg connection to related work, and empirical validation.

**Questions To Authors:**

* How sensitive is APS’s performance to the number of GPUs available for parallel subcalls in real-world deployments?

* Do you think the benefits of APS compared to methods like SoS will be maintained for SOTA-sized models?

* Can you discuss potential challenges in adapting APS to large pre-trained models with much larger context windows?

**Reasons To Accept:**

* Paper introduces a promising inference paradigm APS that generalizes and unifies serialized, parallel, and structured search under a learnable framework.

* Demonstrates strong empirical improvements across multiple axes (latency, context, compute) on a challenging, verifiable reasoning benchmark.

* Offers a well-motivated, end-to-end training approach combining supervision and reinforcement learning that advances the field (while being mindful of scalable deployment and mitigating latency bottlenecks).

* Provides comprehensive experiments and ablations that support key contributions.

**Reasons To Reject:**

* Tested model selection is limited. The provided evidence supports the claims, but I would have liked to see how APS performs with a slightly larger model to get a sense for scaling behavior.

---

> ### Author Response · Authors · 2025-06-03
> **Response**
>
> We sincerely appreciate the reviewer's comments and recognition of our approach.
>
> **Larger models.** Thank you for the suggestion. Following this, we additionally trained APS and the baseline with a 600M-parameter model during the rebuttal. We kindly refer you to the general response for the results. In summary, APS continues to significantly outperform the SoS+ baseline with the 600M-parameter model, especially as total compute increases.
>
> **Performance vs GPUs.** APS’s accuracy remains unchanged with different GPU counts, as the reasoning process itself is unaffected. We refer the reviewer to the general response for variations in latency when we have varying number of GPUs available.
>
> **Performance with SOTA-sized models.** Our rebuttal experiments with the 600M-parameter model, detailed in response to the previous comment, show that APS scales positively with model size and continues to outperform the SoS+ baseline. Note that this is of similar size to representative pretrained models such as Qwen-2.5-0.5B.
>
> This suggests its benefits are likely to persist at larger scales. Due to compute constraints, we couldn’t evaluate billion-parameter models during the rebuttal phase, but as noted in the paper, generalizing APS to larger pretrained LLMs and across broader applications remains an important and promising direction for future work.
>
> **Potential challenges in adapting to large pre-trained models with larger context.** We thank the reviewer for raising this important point. Adapting APS to large pre-trained models with extended context windows introduces two key challenges. First, the increased context length leads to larger KV caches, and when combined with APS’s multi-threaded execution, this results in substantial GPU memory overhead. Efficient resource management becomes critical to ensure that memory constraints do not hinder performance.
>
> Second, integrating APS into distributed serving systems is non-trivial. Large models are typically deployed with tensor or pipeline parallelism, where parameters are split across multiple GPUs or servers. In contrast, APS leverages data parallelism to balance workloads across decoding threads. Combining APS’s dynamic multi-threaded inference with existing parallelism strategies requires careful coordination of tensor, pipeline, and data parallelism, along with thoughtful placement of model parameters across devices to maintain optimal performance. As a promising future direction, we aim to leverage powerful modern serving frameworks such as SGLang and vLLM to address these integration challenges and enable scalable deployment.

---

> > ### Comment · Reviewer_RZt7 · 2025-06-05
> > **Rebuttal Acknowledgement**
> >
> > Thank you for answering my questions and strengthening your work with further experiments. I will maintain my favorable scores. This submission is a prime example of excellent work I would like to see published and highlighted at CoLM.

---

### Official Review · Reviewer_oCwp · 2025-05-20

**Rating:** 6
**Confidence:** 3
**Ethics Flag:** 1

**Summary:**

This work provides another level of test-time compute scaling approach by "multi-threading" at the inference time. It also demonstrates an improved performance with a fixed context window and at an equivalent latency on the Countdown task.

**Questions To Authors:**

Q1: Please refer to the Results comparison section.

Q2: Has this approach been applied to a more powerful model with a longer effective context window length? I feel like this can be an important add up.

Q3:  For tasks that rely more on context window size, the orange curve shown in Figure (3) b can go up as the context window size goes up. Would APS also gain a higher improvement space (for now it saturates with about 2560 but would it saturate at a later stage like 8000+ for a task that relies more on context window size)?

**Reasons To Accept:**

1. The method is effective (reflected as an accuracy boost) on the chosen task.
2. The main idea of "multi-threading" makes sense in this scaling setting.
3. This is an approach to use context windows more effectively.

**Reasons To Reject:**

1. Empirical experiments are weak. The results are demonstrated on one Llama2 model and one specific task Countdown. Plus, the Countdown task given by the author -four candidate numbers with one target number- has a countable exploration space (ignoring partial results and illegal arithmetic operations the space shrinks even more). The approach is effective. I believe it will work on some other task however I doubt if the approach will still have such a big performance boost.

2. Results comparison. What is the prompt template used for the baseline comparison? For the baseline (non-multi-threading) approach, does the work include more information (like increased shot numbers) as the context length increases? I assume the work does but how is this done?

3. Novelty. The idea is fairly straightforward and the approach implemented is quite standard.

---

> ### Author Response · Authors · 2025-06-03
> **Response**
>
> We would like to thank you for the insightful feedback on our work. Below are our clarifications and responses to your comments:
>
> **Empirical experiments and generalization.** In our experiments, we intentionally follow the methodological choices of the Stream of Search (SoS) study (Gandhi et al., 2024; COLM 2024 oral presentation), which centers on the Countdown task due to its structured reasoning challenge, high branching factor, and ease of solution verification. As in SoS, focusing on a controlled yet challenging domain enables us to rigorously evaluate across accuracy, latency, and scalability, providing clear insights into the effectiveness of our proposed approach.
>
> Regarding generalization, the core innovation of our method—adaptive task decomposition and parallelization optimized via reinforcement learning—is naturally suited to more domain-general structured reasoning tasks beyond Countdown, such as puzzle solving and math reasoning. While it is not feasible to include results on benchmarks like GSM8K and HumanEval within the rebuttal period, we plan to explore applying APS to these domains as a next step. We will revise the discussion section to better highlight the generalizability of SoS and outline future directions for extending it to a wider range of tasks.
>
>
> **Prompt template.** Since we performed supervised training on our model from scratch, there is no prompt in the form of textual descriptions for the task. During inference time, the model is asked to perform completion following the prefix for both APS and the baseline. We list the prefix for APS and the baseline below, using the task “constructing 79 from [45, 41, 94, 82]” as an example:
>
> Prefix (APS): `Moving to Node #0\nCurrent State: 79:[45, 41, 94, 82], Operations: []`
> Prefix (Baseline): `Current State: 79:[45, 41, 94, 82], Operations: []`
>
> **Budget conditioning.** For the baseline, length-controlled models were trained by conditioning on context window size. Training samples are binned by length in 512-token increments, and this bin size is used as a conditioning signal. During inference, specifying a bin size controls the generated token count. For example, if the sample length or expected output is between 1537 and 2048 tokens, we prepend "Token Budget: 2048" to the prompt during training and inference. No other additional information, such as increased shot numbers, is included as context length increases. For instance, the SoS+ model in Fig. 3(a) uses a 4096 token budget conditioning, and  Fig. 3(b) demonstrates the effect of token budget conditioning of 1024 and 4096.
>
> **The approach is straightforward and standard.** We see the simplicity of our approach as a strength rather than a limitation. The use of  'spawn' and 'join' operations, inspired by multithreading in operating systems, is intentionally straightforward and forms a clear foundation for scalable parallel reasoning.
>
> While there are prior works exploring parallelized decoding (e.g., self-consistency sampling), **we are among the first to adapt and apply multithreading principles to enable language models to orchestrate parallel reasoning end-to-end**. This approach offers an elegant and effective solution to the limitations of serial decoding in LMs (e.g., latency, context window bottlenecks). We demonstrate that LMs can learn this complex orchestration through a combination of supervised and reinforcement learning, leading to significant improvements in reasoning efficiency and scalability.
>
> We would greatly appreciate citations for specific prior work that renders our approach “standard.” **To the best of our knowledge, few existing work has demonstrated this kind of dynamic, learned task decomposition combined with parallel execution in language models**. We believe our method is a pioneering step toward enabling language models to perform structured reasoning in a more scalable and efficient manner.
>
> **More powerful models & longer context.** We trained a more powerful model with ~600M parameters and found that our method can continue to scale with increased model size, as detailed in our general response. This model size is already larger than some of the pretrained models (e.g., Qwen-2.5-0.5B). APS continues to significantly outperform the SoS+ baseline in this regime.
>
> Note that one strength of APS is that it could reduce the requirements on the context length by exploring multiple directions in parallel. This makes the used context much less compared to the baseline. We demonstrated that the preferred way (as chosen by our end-to-end RL) to scale APS is through increasing the number of child threads, rather than the sequential search. This is shown in Fig. 6: For APS, RL scales the number of total tokens from 10k to 19k, achieving 7% performance boost. In contrast for our baseline, RL only scales the number of tokens from 2.2k to 2.7k, achieving 2.5% performance boost.

---

> > ### Author Response · Authors · 2025-06-05
> > **Response (cont.)**
> >
> > Thank you once again for your thoughtful remarks. We have made several updates during the discussion period, as outlined below.
> >
> > In addition to the Llama-class models used in our original submission, we have extended our experiments to include **Qwen 2.5-1.5B**, a pretrained model roughly 7× larger than our original Llama 2 200M. We fine-tuned Qwen 2.5-1.5B on the same SoS+ and APS training data used previously. As shown in the general response, **APS continues to significantly outperform the SoS+ baseline (80.2% vs. 57.5%)**, demonstrating that the improvements from our method do **not depend on model class (Llama vs. Qwen), size (200M vs. 1.5B), or pretraining status** (from-scratch vs. pretrained).

---

> > ### Comment · Reviewer_oCwp · 2025-06-05
> >
> > I would like to thank the author's effort in addressing my questions. To clarify, by "The approach is straightforward and standard.", I meant multithreading ideas are straightforward and are applied widely. But this work's adaptation into the specified realm is indeed an innovation. The response related to budget conditioning does make things clearer. Most importantly, the expanded experiments solidify the work tremendously.  I am not sure if my Q3 (task-related) is answered specifically. For now, I am willing to raise my score for this paper.

---

> > > ### Author Response · Authors · 2025-06-11
> > > **Response for Q3**
> > >
> > > Thank you for your positive feedback and for raising the score! To specifically address Q3 (**tasks with a longer context**), we conducted additional experiments on the more challenging version of the Countdown task, with 5 input numbers (compared to the standard 4). **This inherently requires much larger context windows** to find a solution than the 4-number task studied in the main paper, **as the search space grows by 40x**. We trained both APS and SoS+ (200M) models for the 5-number task with an 8K-token context.
> > >
> > > As shown in the attached figure (https://imgur.com/a/C4kInUs), APS continues to benefit from longer contexts, **with performance improving up to ~6K tokens** for all 3, 6, and 10 child threads, demonstrating a clear delay in saturation. Notably, APS outperforms the best SoS+ baseline beyond 3.5K tokens, **achieving substantial gains of 7% and 11% at 8K**. This confirms that APS can better leverage extended context through parallelized exploration.

---

### Author Response · Authors · 2025-06-03
**General Response to All Reviewers**

We would like to express our appreciation for the work that the reviewers and organizing members have put in, and thank all reviewers for their valuable feedback. Below we address some of the common questions from the reviewers:

**Larger models (Reviewer oCwp, RZt7, VxmG).** We additionally trained APS and the baseline with a 600M-parameter model during the rebuttal. Results are presented in the following table that mirrors the "Accuracy vs. Avg Total Compute (Tokens)" analysis from our original Figure 3(a), using total compute partitioned into bins. These results demonstrate positive scaling behavior: both APS and the SoS+ baseline generally achieve improved performance with the larger model size across various compute levels. Importantly, APS continues to significantly outperform the SoS+ baseline with the 600M-parameter model, especially as total compute increases.

| **Method**  \ **Avg Total Compute**        | **≈3 k** | **≈5.8 k** | **≈8 k** | **≈10 k** | **≈12 k** | **≈14.5 k** | **≈17.5 k** | **≈20.5 k** |
| ----------------------- | -------- | ---------- | -------- | --------- | --------- | ----------- | ----------- | ----------- |
| **SoS+ Pass@N (200 M)** | 57.3 %   | 63.3 %     | 64.4 %   | —         | 65.7 %    | 66.5 %      | 67.5 %      | 67.8 %      |
| **APS (200 M)**         | 53.5 %   | 61.5 %     | 65.3 %   | 67.9 %    | 69.8 %    | 75.2 %      | 76.9 %      | 80.1 %      |
| **SoS+ Pass@N (600 M)** | 58.1 %   | 63.2 %     | 65.1 %   | —         | 66.2 %    | 67.0 %      | 67.9 %      | 68.2 %      |
| **APS (600 M)**         | 55.5 %   | 62.2 %     | 66.7 %   | 70.0 %    | 73.0 %    | 78.0 %      | 78.0 %      | 83.2 %      |

**Latency when the number of GPUs change (Reviewer RZt7, VxmG).** We evaluated APS using 1 parent thread and 7 child threads under varying GPU configurations. Our experiments, conducted with 1 parent thread and 7 child threads, show that latency increases as GPU count decreases: latency rose by 10% when reducing from 8 → 4 GPUs (4888 → 5436 ms), and by 22% when further reducing from 4 → 2 GPUs (5436 → 6931 ms). When the number of spawned threads exceeds available GPUs, latency increases due to resource sharing. However, our inference engine will batch the decoding process, softening the increase, since decoding two threads together is not twice as slow as decoding the two threads one by one. Higher GPU utilization can be achieved with batched execution, which allows the hardware to process multiple threads more efficiently. As a result, the system maintains reasonable performance, even when threads outnumber GPUs.

| GPUs | Latency&nbsp;(ms) |
|:---:|---:|
| 8 | 4887.555 |
| 4 | 5435.717 |
| 2 | 6930.694 |

Please note that there are some more detailed updates and clarifications in our responses to each reviewer that may not be fully captured in this general response. We kindly request you to refer to our individual responses for a comprehensive understanding of all the changes and additions we plan to incorporate in the revised paper.

### Update to the general rebuttal

**More diverse model classes/sizes and pretrained LLMs.** We trained APS on models with different classes and different sizes. In addition to the llama class of models that we trained from scratch in our original manuscript and the Llama 2 600M model trained in the rebuttal period, we fine-tuned a series of Qwen models by leveraging the pretrained Qwen 2.5-1.5B. We evaluate our model after training with SoS+ and APS training data that we used to train the Llama models in our work, both for ~4000 supervised training steps, respectively. Both settings use the inference setting that leads to the best accuracy in our previous experiments (i.e., with maximum budget/child thread conditioning).

| Setting | Accuracy (Llama 2 200M from Fig. 6) | Accuracy (Qwen 2.5-1.5B) |
|:---:|---:|---:|
| SoS+ | 57.4% | 57.5% |
| APS | **83.2%** | **80.2%** |

This is our initial run with only ~10% in-domain training samples seen compared to the Llama runs from scratch, so further hyperparameter tuning (e.g., adjusting the number of SFT steps) is expected to yield even better performance. Consistent with the trend observed on the Llama 2 backbone, our APS method significantly outperforms SoS+ on Qwen 2.5-1.5B. **These results demonstrate that the performance gains from our approach are not tied to a specific model family (e.g., Llama vs. Qwen), model size (e.g., 200M vs. 1.5B), or whether the model has been pretrained.**

---

### Decision · Program_Chairs · 2025-07-08

**Decision:**

Accept

**Comment:**

This paper proposes Adaptive Parallel Search (APS), a framework that enables language models to adaptively allocate compute between parallel and serial inference. The proposed APS outperforms baselines on multiple dimensions, including latency, context window length, and compute budget. All reviews are leaning towards accepting this submission. However, there are concerns regarding the generalizability of the proposed framework, as all experiments are solely conducted on the Countdown task.